# Fidelity of family centered care model to early disability diagnosis and rehabilitation in the United Arab Emirates

**Maxwell Peprah Opoku**[1]*, **Emma Pearson**[2,3], **Hala Elhoweris**[1], **Najwa Alhosani**[2], **Ashraf Mustafa**[1], **Maria Efstratopoulou**[1], **Rachael Takriti**[2]

1 Department of Special and Gifted Education, United Arab Emirates University, Al Ain, United Arab Emirates, 2 Curriculum & Methods of Instruction, United Arab Emirates University, Al Ain, United Arab Emirates, 3 Froebel Department of Primary and Early Childhood Education, Maynooth University, Maynooth, Ireland

* maxwell.p@uaeu.ac.ae

## Abstract

### Background

The role of parents in supporting early intervention for young children with disabilities is critical. Indeed, models of family centered care (FCC), which emphasis strong partnerships between health professionals and families in disability health services delivery are now widely associated with best practice. While FCC is consistently argued to be an appropriate model for disability service delivery, its utilization is limited primarily to Western countries such as Australia and the United States. Countries such as the United Arab Emirates (UAE) have prioritized early childhood development and are thus in search of best practices for delivery of early intervention for children and their families.

### Objective

The aim of this study was to explore the appropriateness of the FCC model in disability service delivery in the UAE. This study was conducted from the perspectives of health professionals who are involved in disability diagnosis, referral and ongoing support for families and children with disabilities.

### Method

A total of 150 health professionals were recruited from health facilities, rehabilitation centers and schools in the Emirates of Abu Dhabi. The 27-item Measure of Process of Care for Service Providers (MPOC-SP) was used for data collection. The data were subjected to confirmatory factor analysis to confirm applicability of the model to this context. Multivariate analysis of variance and moderation analysis were also conducted, to ascertain the relationship between participants' satisfaction levels with their ability to diagnose, refer and provide on-going support and their likelihood of practicing key components of FCC.

**Data Availability Statement:** The datasets generated and/or analysed during the current study are not publicly available due ethical restrictions. Data contain potentially identifying information.

Data are available from the Social Science Ethics Review Committee at United Arab Emirates University (contact via research.office@uaeu.ac.ae) for researchers who meet the criteria for access to confidential data.

**Funding:** This study was financially supported by the Abu Dhabi Early Childhood Authority. There was no additional external funding received for this study. The funders had no role in study design, data collection and analysis, decision to publish, or preparation of the manuscript. The specific roles of these authors are articulated in the 'author contributions' section.

**Competing interests:** The authors have declared that no competing interests exist.

## Result

Computation of confirmatory factor analysis provided support for applicability of the MPOC-SP in the UAE context. Further inspection showed moderate to large correlations between the four components of FCC measured by MPOC-SP, providing further support for utilization of FCC in disability health service delivery in the UAE.

## Conclusion

The study concludes with a call to policymakers in the UAE to consider developing disability health policy based on key components of FCC. This could be supplemented by development of training modules on FCC to upskill health professionals involved in disability diagnosis and rehabilitation.

## Introduction

Family centered care (FCC) has been discussed widely as a useful model to guide healthcare and disability service delivery and, in particular, early intervention [1–7]. FCC has been defined as effective partnership between health service providers and families on all aspects of health delivery, including planning, delivery and evaluation processes [8–10]. According to Dunst and Trivette [8] and Rosenbaum et al. [7], the implementation of FCC should cover all aspects of child development as the needs of children with disabilities intersect with other areas such as healthcare, education, livelihood and empowerment. According to this model, effective health systems consider inputs from family in deciding the health care needs as well as the communal services which could optimize children's development [1, 7]. Indeed, the parents spend more time with their children than any other person, making them experts with insight knowledge about their children [11] which could be capitalized on and considered in health services delivery. There is growing evidence to the effect that family involvement or consultation at all stages of the diagnosis or rehabilitation has positive impact on child development [1] and better support practices [12, 13]. This lends support for health service providers to consider adopting FCC as a model in delivery of diagnosis and rehabilitation services to children with disabilities and their families.

There are widespread scholarly discussions on tenets of FCC which could be adopted by health systems. For instance, in a review study, Kokorelias et al. [14] summarized literature on FCC and proposed a universal model of FCC which incorporates four indicators: consideration of family context; patient, family and care provider collaboration; illness specific education, and dedicated policies and procedures. In his conceptualization of FCC, Rosenbaum et al. [7] outlined five guiding principles: family involvement in decision-making, responsibility for care, treating families with respect, family needs and encouragement of family involvement. In a review of literature, King and Chiarello [9] summarized models such as relational goal-oriented model, collaborative practice model, participation-based therapy model and coaching model to guide the implementation of FCC in health systems. Consensus on the appropriateness of FCC is growing, however there seems to be a lack of agreement regarding the ideal model which could be adopted by a given context to guide disability service delivery [1, 3, 4, 6, 9]. Nonetheless, health systems could experiment and develop their disability health services around one of the models of FCC.

In line with this, various measurement tools have been developed to assess health professionals' knowledge or understanding of FCC [14]. For instance, enabling practice scale [15], family empowerment scale [16], client satisfaction questionnaire [17], family support scale [8] and the measure of processes of care (MPOC) [1, 18, 19] has been used to develop understanding of the implementation of FCC. One widely used tools that has received theoretical support from various contexts in studies of perceived implementation of FCC by service providers, is the Measures of Processes of Care for Service Providers (MPOC-SP) [1, 18, 19]. This supports its continued usage to study professionals' perceptions towards adoption of FCC in the delivery of disability health services.

The MPOC-SP consists of four key components that measure the nature of professionals' interactions with caregivers: Treating people with respect, Interpersonal sensitivity, communicating specific information (focused on the individual child's disability) and Providing general information (to support caregivers in working with wider systems and supports) [20, 21]. Previous studies conducted among professionals using the MPOC-SP have shown a particular pattern. For instance, consistently, it has been reported that ratings on Treating people with respect are higher compared with ratings on Providing general information [22–25]. Specifically, in a Finnish study of using MPOC, both parents and service providers rated their experience of FCC as ranging from fair to moderate [24, 26]. While their ratings were high on Treating people with respect, providing general information emerged as the lowest [24, 26]. Similarly, Tang et al. [25], in a Chinese study of professional perceptions towards implementation of FCC, they noted that Treating people with respect was rated highly while Providing general information received the lowest rating.

Differences have also been found between participants on the extent to which they implement FCC [2, 23, 27]. In an Australian study of parents and professionals understanding of FCC, the mean ratings showed that practitioners were high on most of the tenets compared to parents [27]. While ratings were high on Treating people with respect, General information received the lowest ratings. In a US study, McManus et al. [2] reported difference in the ratings of both caregivers and providers. While caregivers and providers ratings were low on providing general information, caregivers' ratings was higher compared to the practices. However, both groups ratings were higher on treating people with respect, with caregivers' mean ratings once again, higher than practitioners. This consistent pattern of findings indicates that health professionals struggle to engage with the wider issues beyond medical treatment and therapies that families encounter in providing support for children with disabilities. Exploration of whether similar patterns exist in non-Western contexts, such as the United Arab Emirates (UAE), could shed light on the universal applicability of concepts associated with FCC.

In the UAE, there is limited evidence on the efficacy and / or applicability of FCC in health service delivery for families raising children with disabilities. The reliability of the family version of the MPOC-SP investigated in Jordan [28], and the practitioner's version has been validated for use in Iran [29], indicating applicability of the framework within Middle eastern contexts. However, the efficacy of the MPOC-SP for measuring FCC among allied health professionals in early intervention for children with disabilities is yet to be studied. The UAE has an advanced health system [30, 31] and has introduced a raft of policies that reflect commitment to creating a conducive environment for persons with disabilities and their families [32–35]. Recent developments include the establishment of Abu Dhabi Early Childhood Authority (ECA), which has begun to establish robust structures for supporting positive early development of children and families. As part of its work, the ECA is concerned with understanding how to strengthen early intervention and health service delivery in Abu Dhabi. The research reported here was thus commissioned, with one of its aims to assess the fidelity of the FCC model in Abu Dhabi. This study also extends previous studies, by considering the relationship

between professionals' satisfaction with early intervention and disability service delivery and the four components of the MPOC-SP (Treating people with respect, Interpersonal sensitivity, communicating specific information and Providing general information). Will MPOC-SP emerge as a valid tool to measure FCC to disability diagnosis and rehabilitation in the UAE context?

1. Are three notable differences between health professionals on their implementation of the four components of the MPOC-SP in the UAE context?

2. What are the moderators of links between satisfaction in ability to diagnose, refer and provide on-going support, and practicing the four components of FCC, as measured by MPOC-SP, in the UAE context?

## Method

### Study participants

The study participants were allied health professionals who were working in health facilities, rehabilitation centres and schools in one out of the seven emirates in the UAE. Abu Dhabi is the national capital of UAE with an estimated population of about three million [36]. Administratively, the Emirate is divided into three regions: Abu Dhabi, Al Dhafra and Al Ain. According to the Ministry of Community Development, Abu Dhabi is home to 20,000 people with disabilities [37].

The survey was also distributed via researcher networks, to rehabilitation centres and schools across Abu Dhabi. Specifically, with support from the Abu Dhabi Department of Health and the Abu Dhabi Early Childhood Authority, surveys were distributed, using the online survey platform Qualtrics, to the full list of allied health professional registered with the Department of Health in Abu Dhabi (N = 2513). The list includes physiotherapists (n = 1649), speech and language therapists (n = 222), psychologists (n = 153), psychiatrists (n = 216) and occupational therapists (n = 273).

Overall, 252 which is 10% allied health professionals in Abu Dhabi gave consent and entered the survey. The sample size was deemed appropriate based on prior computation using *OpenEpi* to estimate the expected sample (https://www.openepi.com/SampleSize/SSPropor.htm). The expected sample was 182 using the following estimate parameters:

$$\text{Confidence interval} : 95\%$$

$$\text{Hypothetical percentage of frequency of outcome} : 15\% +\_5\%$$

During data cleaning, close inspection of the data showed that many of the participants did not complete the full survey. In most instances, participants completed only the demographic information without finishing the MPOC-SP. After deleting the empty entries, 150 valid participants were used in the reporting of this study (see Table 1 for details). While 91% were expatriates, 9% were Emiratis. This is line with data of ratio of expatriates (96%) and citizens working (4%) in the healthcare profession in Abu Dhabi [38].

### Instrument

A three-part survey instrument was used for data collection. The first part covered demographic information of study participants: gender, nationality, area of work, specialization, key roles and responsibilities and working experience. Ratings of participants' satisfaction with

**Table 1. Summary of demographic characteristics of health professionals.**

| Categories | Frequency | Percentage (%) |
|---|---|---|
| **Nationality** | | |
| Expatriate | 136 | 91% |
| Emiratis | 14 | 9% |
| **Gender** | | |
| Male | 63 | 42% |
| Female | 87 | 58% |
| **Place of work** | | |
| Abu Dhabi | 86 | 58% |
| Al Ain | 62 | 42% |
| **Specialization** | | |
| Language and speech pathology | 16 | 11% |
| Occupational health | 29 | 19% |
| Physiotherapy | 64 | 43% |
| Other | 40 | 27% |
| **Working experience** | | |
| Less than 5 years | 22 | 15% |
| 6–10 years | 39 | 26% |
| 11–15 years | 46 | 31% |
| At least 16 years | 43 | 29% |
| **Experience with diagnosis** | | |
| Less than 5 years | 39 | 26% |
| 6–10 years | 42 | 28% |
| 11–15 years | 36 | 24% |
| At least 16 years | 32 | 22% |
| **Qualification** | | |
| Bachelor | 70 | 57% |
| Masters | 39 | 32% |
| PhD | 15 | 12% |
| **Place of training (country)** | | |
| In-country | 35 | 23% |
| Out-of-country | 115 | 77% |
| **Nature of involvement** | | |
| Diagnosis and referral | 10 | 7% |
| Ongoing therapy and support | 66 | 44% |
| All the above | 73 | 49% |
| **Age of diagnosis** | | |
| 0–12 months | 58 | 39% |
| 12–36 months | 34 | 23% |
| 4–10 years | 31 | 21% |
| 11–18 years | 25 | 17% |
| Number of tools for diagnosis | | |
| Only 1 tool | 22 | 21% |
| Two tools | 26 | 25% |
| Three tools | 19 | 18% |
| 4 or more tools | 37 | 36% |

Note: place of training = country where participants were trained

their ability to provide accurate diagnosis, referral and on-going support for children with disabilities were also included in this section.

These comprised four items across three main areas: satisfaction towards diagnosis, satisfaction towards referral and satisfaction towards providing ongoing support to children with disabilities and their families. The instrument was anchored on a 6-point Likert scale with scores ranging from 1 (very satisfied) to 6 (not applicable). A composite mean score of at most 2 was interpreted as high satisfaction among professionals who took part in this study.

The second part comprised the 27-item MPOC-SP [1, 18, 19], which has been widely used and yielded appropriate psychometric properties across diverse contexts. Permission to use the instrument for this study was obtained from the developers. The instrument is made up of four sub-scales [Providing general information, n = 5; Showing interpersonal sensitivity, n = 10; Communicating specific information, n = 3; and Treating people with respect, n = 9] and anchored on a 7-point Likert scale with scores ranging from 0 (not applicable) to 7 (to a very great extent).

The instrument was subjected to content validation from reputable scholars in pediatrics, disability rehabilitation and recognized institutions. Feedback from these institutions were incorporated in the final draft used for data collection from health professionals.

## Procedure

The study and its protocols were approved by Social Science Research Ethics Committee at United Arab Emirates University (ERS_2022_8487) and Abu Dhabi Department of Health (DOH/CVDC/2022/1096).

The data were collected between November 2022 and March 2023. Surveys were distributed in bilingual form, using both Arabic and English. Back-to-back translations were conducted by three researchers, with research team meetings to confirm clarity of items and translation. The final version of the survey was piloted among a small number of allied health professionals to further ensure clarity of items and translation. An online information page outlining the purpose of the study and funding supports was provided to participants as part of the online survey, and entry into the survey was conditional upon participant signing informed consent.

## Data analysis

The data collected via Qualtrics were transferred to Microsoft Excel for cleaning before being transferred to SPSS version 28 for further analysis. Although Kolmogorov-Smirnov and Shapiro-Wilk tests were below the .05, observation of the histograms, box-plots and Q-Q plots supported normality of the data for parametric analyses. According to Pallant [39], the data could be presumed to be reasonably normal. Following this, missing at random test was conducted to check the missing data which yielded a score of below 10%. Following this, missing data were imputed using expectation-maximization algorithm.

To answer research question 1, confirmatory factor analysis (CFA) was used to explore the underlying factor structure of the MPOC-SP scale in a novel context. The scale's appropriateness was assessed using the following indices fit: a chi-square value of less than 5; comparative fit index (CFI) and Tucker–Lewis index (TLI) values of .90 or greater; root mean square error of approximation (RMSEA) and standardized root mean square residual (SRMR) values of between .03-.08; and a regression weight of at least .50 [40–42]. In the event of poorly fit model, modification index was checked to ascertain items whose erroneous correlations might be impacting adversely on the model [40]. The items were either deleted or correlated in the event they are within the same sub-scales [40].

The fidelity of the model was measure from correlation between the sub-scales, with large correlation suggestive of appropriateness of the FCC model in the UAE. Correlations between the latent variables were classified as small (.10–.30), moderate (.31–.50), or large (at least .51) [39].

To answer research question 2, multivariate analysis of variance (MANOVA) was computed. The demographic variables (e.g. gender) were used as independent variables, while the continuous variables were used as dependent variables (providing general information, showing interpersonal sensitivity, communicating specific information, and treating people with respect). There was no serious violation of the following assumptions: normality, linearity, outliers and homogeneity of variance [39]. A Bonferroni-adjusted alpha level of .01 (which is .05 divided by the number of dependent variables) was used as the baseline to determine whether there were differences between the participants [39].

To answer research question 3, Andrew Haye's Process model 1 [43], which is embedded in SPSS, was used for moderation analysis to determine the influence of demographics on the relationship between satisfaction and FCC. The moderators were the demographic variables. The independent variable was satisfaction, while the outcome variable was the four components of FCC. In the imputation of the model, the bootstrap and bias corrected confidence interval were set at 500 and 95%, respectively.

## Results

### Structural validity of MPOC-SP

The 27-item short form MPOC-SP was subjected to CFA to validate the underlying factor structure in an Arab context. The initial computation of CFA showed a poorly fit model with the following indices: chi-square = 4.92 ($CMIN$ = 2638.66/$df$ = 536), $CFI$ = .77, TLI = .71, $RMSEA$ = .14 and $SRMR$ = .06). Observation of the regression weight showed that all the items loaded above .05. Following this, modification indices was assessed to determine whether correlations between the items is having effect on the model. Iterate removal of items improved the model. While three items were removed from Interpersonal Sensitivity sub-scale, four items were removed from the Treating People with Respect sub-scale.

Nineteen items were supported in the current study context with the following improved fit indices: chi-square = 2.62 ($CMIN$ = 375.15/$df$ = 143), $CFI$ = .92, $TLI$ = .90, $RMSEA$ = .10 and $SRMR$ = .06 (Fig 1 summarizes the correlation between items of MPOC). There was moderate to large correlation between the sub-scales.

Computation of reliability using Cronbach Alpha yielded the following result: total MPOC = .94; providing general information = .95; Treating People with Respect = .94; communicating specific information = .83; and interpersonal sensitivity = .92.

### Level of awareness of family centered practice

The overall level of practitioners' awareness of FCC was $M$ = 5.82 ($SD$ = .93) and the sub-scales were as follows: Providing general information, $M$ = 5.39 ($SD$ = 1.46); Treating people with respect, $M$ = 6.16 ($SD$ = .96); Communicating specific information, $M$ = 5.99, $SD$ = .96; and Showing interpersonal sensitivity, $M$ = 5.82 ($SD$ = 1.17).

### Difference between participants on FCC

MANOVA was computed to explore the difference between participants on FCC (see Table 2). Difference was found between participants on place of work on the combined dependent

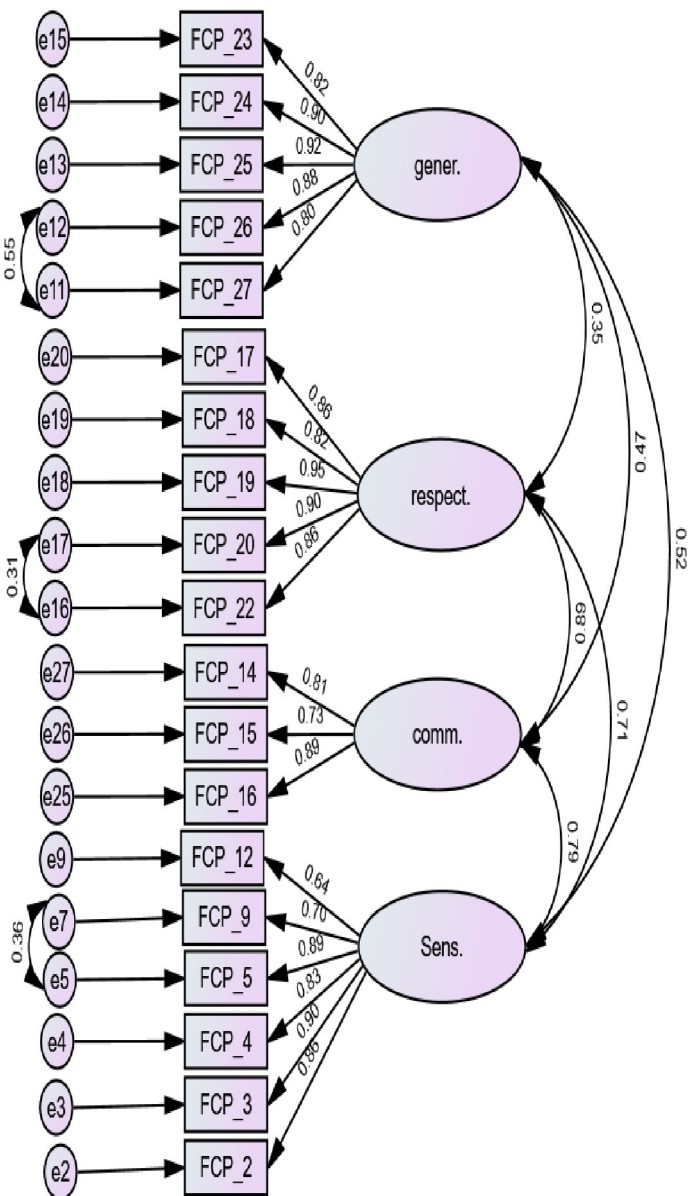

**Fig 1. Summary of confirmatory factor analysis for the Measure of Processes of Care. Note**: gener. = providing general information; respect. = Treating People with Respect; comm. = communicating specific information; sens. = interpersonal sensitivity.

variables only, [$F$ (4, 146) = 4.65, *Wilks' Lambda* = .89, $p$ = .001], with a very large effect size, *partial eta squared* = .12.

Individually, difference was found between participants on Treating people with respect [$F$ (1, 146) = 8.16, $p$ = .005, *partial eta squared* = .05] and Communicating specific information [$F$ (1, 146) = 5.40, $p$ = .02, *partial eta squared* = .04] and interpersonal sensitivity [$F$ (1, 146) = 15.51, $p$ = .10].

Observation of the mean scores showed as follows: Treating people with respect [Abu Dhabi, $M$ = 6.35, $SD$ = .73; Al Ain, $M$ = 5.90, $SD$ = 1.18], Communicating specific information [Abu Dhabi, $M$ = 6.14, $SD$ = .89; Al Ain, $M$ = 5.77, $SD$ = 1.05] and interpersonal sensitivity

**Table 2. Multivariate analysis of variance for comparison between participants.**

| | Wilks' Lambda | MANOVA F | ANOVA F | | | |
| --- | --- | --- | --- | --- | --- | --- |
| | | | **Providing** | **Respect** | **Commun.** | **Sensit.** |
| Nationality | .94 | 2.35 | 6.05* | 3.67 | 5.63* | 1.49 |
| *Effect size* | | .06 | .04 | .06 | .02 | .01 |
| Gender | .99 | .39 | .69 | .003 | .06 | .78 |
| *Effect size* | | | .005 | .001 | .001 | .005 |
| Place of work (Regions in Abu Dhabi) | .89 | 4.65** | .05 | 8.16* | 5.40* | 15.51* |
| *Effect size* | | .12 | .001 | .05 | .04 | .10 |
| Specialization | .89 | 1.47 | .06 | 5.19* | 2.20 | 2.40 |
| *Effect size* | | .04 | .001 | .10 | .04 | .05 |
| Working experience | .88 | 1.58 | 1.34 | 1.70 | 2.17 | 3.45 |
| *Effect size* | | .04 | .03 | .03 | .04 | .07 |
| Experience with disability diagnosis | .95 | .66 | .95 | 1.00 | 1.82 | 1.46 |
| *Effect size* | | .02 | .02 | .02 | .04 | .03 |
| Qualification | .95 | .75 | .54 | .11 | .47 | .02 |
| *Effect size* | | .03 | .009 | .002 | .008 | .001 |
| Place of training (country) | .97 | 1.02 | .37 | .18 | .59 | .50 |
| *Effect size* | | .03 | .002 | .001 | .004 | .003 |
| Nature of involvement | .94 | 1.07 | .63 | 2.12 | 2.23 | 2.49 |
| *Effect size* | | .03 | .009 | .03 | .03 | .03 |
| Age of disability diagnosis | .95 | .63 | .23 | .94 | .88 | 1.70 |
| *Effect size* | | .02 | .005 | .02 | .02 | .03 |
| Number of tools | .88 | 1.03 | 1.56 | .26 | .09 | 1.19 |
| *Effect size* | | .04 | .05 | .008 | .003 | .03 |

**Note**:

*P < .01 based on Bonferroni adjustment; providing. = Providing general information; Treating people with respect; Communicating specific information; Showing interpersonal sensitivity.

[Abu Dhabi, $M$ = 6.13, $SD$ = .87; Al Ain, $M$ = 5.39, $SD$ = 1.41]. It was apparent that those who indicated that they were working in Abu Dhabi Seemed more competent in the adoption of FCC than those working in Al Ain.

On nationality, though there was no difference between participants on the combined dependent variables [$F$ (4, 145) = 2.35, $p$ = .06, *Wilks' Lambda* = .94, large effect size, *partial eta squared* = .06]. However, individually, there were differences between participants on Providing general information [$F$ (1, 148) = 6.05, $p$ = .02, *partial eta squared* = .04] and Communicating specific information [$F$ (1, 148) = 5.63, $p$ = .02, *partial eta squared* = .04]. The mean scores were as follows: Providing general information [Expat, $M$ = 5.49, $SD$ = 1.38; Emiratis, $M$ = 4.49, $SD$ = 1.92] and Communicating specific information [Expat, $M$ = 6.05, $SD$ = .96; Emiratis, $M$ = 5.41, $SD$ = .85].

Moreover, on specialization, though there was no difference between dependent variables on the combined dependent variables [$F$ (4, 142) = 1.47, *Wilks' Lambda* = .89, $p$ = .13, with a moderate effect size, *partial eta squared* = .04], difference was found between participants on Treating people with respect [$F$ (3, 145) = 5.19, $p$ = .002, *partial eta squared* = .10]. Post-hoc comparison using Tukey HSD test showed that those who self-identified as "other" [$M$ = 5.68, $SD$ = 1.30] and language and speech pathology [$M$ = 6.39, $SD$ = .69], occupational health [$M$ = 6.40, $SD$ = .71] and physiotherapy [$M$ = 6.31, $SD$ = .75] who did not differ from each other.

## Relationship between FCC and satisfaction

The three items were used to measure satisfaction of health professionals towards diagnosis, referral and ongoing support. The instrument yielded appropriate reliability score of .72 and a mean of, 2.11(1.01).

Initial calculation of Pearson Correlation coefficient showed no relationship between satisfaction and Treating people with respect [$r$ = -.07, $p$ = .42]. However, there was correlation between satisfaction and other three components (satisfaction and Providing general information, [$r$ = -38, $p$ = .001]; satisfaction and Communicating specific information, [$r$ = -.25, $p$ = .002]; satisfaction and Showing interpersonal sensitivity, [$r$ = -.34, $p$ = .001] (note: satisfaction anchored from 1 (very satisfied) to 6 (not applicable) while FCC was anchored from 1 (not applicable) to 7 (to a very great extent).

Andrew Haye's Model Method 1 was used to compute moderation analyses to explore the relationship between satisfaction and the four sub-scales of FCC (Table 3). Satisfaction was operationalized as an independent variable to ascertain its influence on adoption of FCC. Also, the demographic variables were used as moderators to understand its impact on the relationship between the dependent and the independent variable. The results showed that only two demographic variables significantly moderated the relationship between satisfaction and two sub-scales of FCC.

For instance, gender moderated the relationship between satisfaction with diagnosis and Communicating specific information, [$beta$ = .51, 95% $CI$ (.21, .81), $t$ = 3.39, $p$ = .0009]. Both gender [$beta$ = -3.13, 95% $CI$ (-5.20, -1.06), $t$ = -2.99, $p$ = .003] and satisfaction [$beta$ = -1.06, 95% CI (-1.56, -.56), $t$ = -4.18, $p$ = .0001] impacted positively on Communicating Specific information. On gender, individually, among male participants, a significant relationship was found between satisfaction and Communicating specific information, [$beta$ = -.55, 95% $CI$ (-.78, -.31), $t$ = -4.65, $p$ = .001]. Conversely, for female participants, no relationship was found between satisfaction and Communicating specific information.

Second, there was also a moderation effect of place of work on the relationship between satisfaction and Showing interpersonal sensitivity, [$beta$ = -.82, 95% $CI$ (-1.51, -13), $t$ = -2.34, $p$ = .02]. Interestingly, both place of work [$beta$ = .36, 95% $CI$ [-4.43, 5.16], $t$ = .15, $p$ = .88] and satisfaction [$beta$ = .27, 95% $CI$ [-.73, 1.27], $t$ = .59, $p$ = .59] did not have direct effect on Showing interpersonal sensitivity. Individually, when place of work was considered separately, among participants working in Abu Dhabi [$beta$ = -.55, 95% $CI$ (-.96, -.13), $t$ = -2.58, $p$ = .01] and Al Ain [$beta$ = -1.36, 95% $CI$ [-1.92, -.81], $t$ = -4.89, $p$ = .001], differences were found between satisfaction and Showing interpersonal sensitivity.

## Discussion

This study aimed to investigate the efficacy of the FCC model in UAE context, by exploring of allied health professionals' implementation of FCC in the UAE. While computation of CFA helped to assess appropriateness of the MPOC-SP tool, there was also high correlation between the latent variables. Based on our proposition that moderate to large correlations between the latent variables suggest appropriateness of the FCC model, it can be argued that the four indicators measured by the MPOC-SP (Providing general information, Communicating specific information, Treating people with respect and Showing interpersonal sensitivity) could be considered in future health service delivery policy development in the UAE. Previous studies conducted in UAE have indicated barriers faced by families in their search for appropriate diagnosis, referral and on-going supports for their children with disabilities [44–48]. Evidence from a range of other contexts indicates that FCC provides an effective model for achieving positive outcomes for children with disabilities and their families [1, 12, 13]. Findings from

**Table 3. Moderators of the relationship between satisfaction and FCC.**

| | Beta | Standard error | t | p | Confidence interval | |
|---|---|---|---|---|---|---|
| | | | | | Lower | Upper |
| **Nationality** | | | | | | |
| Providing general information | -1.08 | .74 | -1.46 | .15 | -2.55 | .38 |
| Communicating specific information | .13 | .31 | .43 | .67 | -.48 | .75 |
| Treating people with respect | .57 | .53 | 1.06 | .29 | -.49 | 1.62 |
| Showing interpersonal sensitivity | .63 | .74 | .85 | .40 | -.84 | 2.10 |
| **Gender** | | | | | | |
| Providing general information | .45 | .38 | 1.18 | .24 | -.30 | 1.19 |
| Communicating specific information | .51 | .15 | 3.39 | .0009** | .21 | .81 |
| Treating people with respect | .47 | .27 | 1.78 | .08 | -.05 | 1.00 |
| Showing interpersonal sensitivity | .62 | .37 | 1.68 | .10 | -.11 | 1.35 |
| **Place of work** | | | | | | |
| Providing general information | -.23 | .39 | -.59 | .55 | -1.00 | .54 |
| Communicating specific information | .02 | .16 | .10 | .92 | -.30 | .33 |
| Treating people with respect | .15 | .27 | .54 | .59 | -.38 | .67 |
| Showing interpersonal sensitivity | -.82 | .35 | -2.34 | .02* | -1.51 | -.13 |
| **Specialization** | | | | | | |
| Providing general information | .07 | .21 | .36 | .72 | -.33 | .48 |
| Communicating specific information | .08 | .08 | .97 | .33 | -.08 | .25 |
| Treating people with respect | .12 | .14 | .84 | .40 | -.16 | .40 |
| Showing interpersonal sensitivity | .02 | .19 | .12 | .90 | -.36 | .41 |
| **Working experience** | | | | | | |
| Providing general information | .001 | .17 | .01 | .99 | -.34 | .35 |
| Communicating specific information | .03 | .07 | .38 | .71 | -.12 | .17 |
| Treating people with respect | -.03 | .12 | -.28 | .78 | -.28 | .21 |
| Showing interpersonal sensitivity | -.17 | .17 | -1.01 | .31 | -.51 | .16 |
| **Experience with disability diagnosis** | | | | | | |
| Providing general information | .28 | .21 | 1.35 | .18 | -.13 | .68 |
| Communicating specific information | .11 | .09 | 1.26 | .21 | -.06 | .28 |
| Treating people with respect | -.01 | .15 | -.07 | .94 | -.30 | .28 |
| Showing interpersonal sensitivity | .17 | .20 | .86 | .39 | -.22 | .56 |
| **Qualification** | | | | | | |
| Providing general information | -.16 | .35 | -.46 | .64 | -.85 | .53 |
| Communicating specific information | -.17 | .14 | -1.20 | .23 | -.45 | .11 |
| Treating people with respect | -14 | .25 | -.58 | .57 | -.63 | .35 |
| Showing interpersonal sensitivity | .22 | .35 | .62 | .54 | -.47 | .91 |
| **Place of training (country)** | | | | | | |
| Providing general information | .85 | .79 | 1.08 | .28 | -.70 | 2.40 |
| Communicating specific information | .06 | .33 | .18 | .86 | -.59 | .71 |
| Treating people with respect | .02 | .56 | .04 | .97 | -1.08 | 1.13 |
| Showing interpersonal sensitivity | .19 | .77 | .24 | .81 | -1.35 | 1.71 |
| **Nature of involvement** | | | | | | |
| Providing general information | -.23 | .40 | -.57 | .57 | -1.02 | .57 |
| Communicating specific information | .07 | .17 | .39 | .69 | -.26 | .40 |
| Treating people with respect | .05 | .28 | .19 | .85 | -.51 | .62 |
| Showing interpersonal sensitivity | .44 | .39 | 1.12 | .27 | -.34 | 1.20 |
| **Age of disability diagnosis** | | | | | | |

*(Continued)*

**Table 3.** (Continued)

| | Beta | Standard error | t | p | Confidence interval | |
|---|---|---|---|---|---|---|
| | | | | | Lower | Upper |
| Providing general information | .001 | .15 | -.0001 | 1.00 | -.30 | .30 |
| Communicating specific information | .03 | .06 | .46 | .65 | -.10 | .15 |
| Treating people with respect | .05 | .11 | .50 | .62 | -.16 | .27 |
| Showing interpersonal sensitivity | .16 | .15 | 1.09 | .28 | -.13 | .45 |
| **Number of tools** | | | | | | |
| Providing general information | .30 | .25 | 1.21 | .23 | -.19 | .80 |
| Communicating specific information | -.08 | .08 | -.91 | .36 | -.26 | .10 |
| Treating people with respect | -.11 | .14 | -.76 | .45 | -.38 | .17 |
| Showing interpersonal sensitivity | -.19 | .19 | -.99 | .32 | -.56 | .19 |

*$p < .05$

**$p < .01$

this study support the applicability of FCC approaches in the UAE. Approaches to early intervention informed by the four components measured in this study could result in strengthened models of FCC.

Overall, the study participants rated their current implementation of FCC, as measured across the four components, as occurring to a "fairly great extent". However, individual ratings on each component ranged from "a great extent" on Treating people with respect, to a "fairly great extent" on the other components. While participants rated themselves highly on Treating people with respect, the lowest rating was found on Providing general information. These dynamics are consistent with previous studies using MPOC-SP, which have also reported high ratings on Treating people with respect and least on Providing general information [22, 23, 25, 26]. Items that constitute the Providing general information component include reference to providing information about broader concerns that parents might encounter associated with caring for a child with disability (financial concerns, genetic counselling), as well as linking parents to other service providers that might offer supports, including community-based services. These aspects of FCC do not commonly form part of professional training programmes, despite them being widely recognized as constituting and important aspect of FCC [18]. This probably lend support for health training institutions to integrate aspect of FCC in pre-service training and professional development curriculum.

The relationship between professionals' levels of satisfaction around being able to provide diagnosis, referral and on-going supports, and implementation of FCC components was also assessed. Previous studies have consistently reported relationships between health professionals' satisfaction and work-related performance or recognition [49–51]. The current study supported such findings, as a positive relationship was found between levels of satisfaction and the three constructs, communicating specific information, providing general information and Showing interpersonal sensitivity. This finding suggests that professionals who feel confident that they are able to provide good quality services are more likely to incorporate an FCC approach in their practice.

One demographic variable that was revealed as moderating links between levels of satisfaction and implementation of FCC components was place of work. Computation of MANOVA and moderation showed differences between participants. While MANOVA showed differences between participants on three sub-scales (communicating specific information, treating providing with respect and showing interpersonal sensitivity), moderation analysis showed

difference between participants on one sub-scale, Showing interpersonal sensitivity. However, on both computations, it was apparent that those working in Abu Dhabi appeared to score higher than those working in Al Ain. This study partly agrees with previous studies which found difference between allied health professionals based on place of work (urban vs. rural workers) [23]. Though both areas may not be considered as urban areas, Abu Dhabi is the national capital with a high population, and more advanced facilities for supporting children with disabilities. While further research is needed to better understand the different professional experiences of health professionals in the regions of Abu Dhabi and town of Al Ain, this finding indicates, first, that disaggregating data to account for geographical distinctions is important. Second, it could be that with the greater spread of early intervention supports in Abu Dhabi, there are more opportunities for in-service training and strengthened policies around the implementation of FCC across some facilities. Ensuring that all registered professional have access to in-service training to support FCC is important for implementation of FCC practices across institutions.

Gender also moderated the relationship between satisfaction and Communicating specific information. Although male and female professionals did not differ on satisfaction, they differed on communicating specific information. Specifically, male professionals appear to be more likely to self-report that they communicate specific to parents about their child's disability than female health workers. Interestingly, similar research conducted in Italy to study perceived implementation of FCC also found that male participants reported higher levels of FCC in their practice. The authors of the Italian study suggest that this could be linked to higher levels of optimism among male professionals, or lower expectations regarding family involvement [52]. These explanations would also be applicable in the UAE context, where female health workers may be less confident in their own abilities. Once again, this finding implies the importance of on-going professional training and support, to ensure that all professionals are confident in providing FCC.

## Study limitations

This study reported here cannot be generalized because of a number of study limitations. First, the study relied on self-reported account of health professionals working in one out of the seven Emirates in the UAE. Self-reported assessment is prone to be response bias and as such, future use qualitative interviews to develop in-depth insight into the experiences of health professionals when it comes to the implementation of FCC. Additionally, the research team did not have direct encounter with the study participants. Second, the study was conducted in one out of the seven Emirates. This suggests that interpretation of the study findings could be limited to the Emirates of Abu Dhabi. Future studies may consider recruiting participants across the country to compare experiences. Third, a large number of entries were deleted due to non-completion of the survey. This could be a result of the length of the survey or participants were uncomfortable completing an online survey. Virtual data collection was appropriate due to outbreak of COVID-19. Future study may use paper-based approach for data collection from allied health professionals within the UAE or similar context to compare with the findings reported in this study.

## Conclusion and policy implications

The current study explored the perceived competence of health professionals towards implementation of FCC in the UAE. The findings provided theoretical support for MPOC-SP and the FCC model in the current study context. As early intervention is the next frontier in health and child development in the UAE, the findings reported in this study may be considered in

future policy development. In particular, policymakers may consider developing early intervention policy based on the components of FCC. This could be followed with development of training modules based on the tenets of FCC which could be used to train health professionals to equip them with knowledge and skills.

## Acknowledgments

Our heartfelt gratitude goes to all the healthcare workers who took part in this study.

## Author Contributions

**Conceptualization:** Maxwell Peprah Opoku, Emma Pearson, Hala Elhoweris, Najwa Alhosani, Ashraf Mustafa, Maria Efstratopoulou, Rachael Takriti.

**Data curation:** Maxwell Peprah Opoku, Emma Pearson.

**Formal analysis:** Maxwell Peprah Opoku, Emma Pearson.

**Investigation:** Maxwell Peprah Opoku, Emma Pearson.

**Methodology:** Maxwell Peprah Opoku, Emma Pearson, Hala Elhoweris, Najwa Alhosani, Ashraf Mustafa, Maria Efstratopoulou, Rachael Takriti.

**Project administration:** Emma Pearson.

**Writing – original draft:** Maxwell Peprah Opoku, Emma Pearson, Hala Elhoweris, Najwa Alhosani, Ashraf Mustafa, Maria Efstratopoulou, Rachael Takriti.

**Writing – review & editing:** Maxwell Peprah Opoku, Emma Pearson, Hala Elhoweris, Najwa Alhosani, Ashraf Mustafa, Maria Efstratopoulou, Rachael Takriti.

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
