## [Decision Letter · Decision Letter 0]

20 Feb 2024

PONE-D-24-03770Fidelity of family centered care model to early disability diagnosis and rehabilitation in the United Arab EmiratesPLOS ONE

Dear Dr. Opoku,

Thank you for submitting your manuscript to PLOS ONE. After careful consideration, we feel that it has merit but does not fully meet PLOS ONE’s publication criteria as it currently stands. Therefore, we invite you to submit a revised version of the manuscript that addresses the points raised during the review process.

We look forward to receiving your revised manuscript.

Kind regards,

Amin Nakhostin-Ansari

Academic Editor

PLOS ONE

Journal Requirements:

“The research reported here was supported by funding provided by the Abu Dhabi Early Childhood Authority”

Reviewers' comments:

Reviewer's Responses to Questions

**Comments to the Author**

1. Is the manuscript technically sound, and do the data support the conclusions?

Reviewer #1: Yes

Reviewer #2: Partly

2. Has the statistical analysis been performed appropriately and rigorously? 

Reviewer #1: Yes

Reviewer #2: I Don't Know

3. Have the authors made all data underlying the findings in their manuscript fully available?

Reviewer #1: Yes

Reviewer #2: Yes

4. Is the manuscript presented in an intelligible fashion and written in standard English?

Reviewer #1: Yes

Reviewer #2: Yes

5. Review Comments to the Author

Reviewer #1: I am greatly happy to review this article Fidelity of family centered care model to early disability diagnosis and rehabilitation in the United Arab Emirates. There are a few points that need to be addressed. The study is good and very helpful for early disability management and making health policies.

1- Abu Dhabi is the national capital of UAE with an estimated population of about three million - Please give reference.

2- 10-15% response rate is acceptable in any good study, 150 response is very low.

3- How did authors calculated sample size?

4- Only 9% of respondent were Emiratis, what is the reason, what is the percentage of Emiratis in Healthcare Professionals?

5- 43% response were from Physiotherapist, whereas children with disabilities are mostly in close relation with Occupational Therapist, numbers of OT were few in health profession, or what may be the reason?

6- What do you mean ‘Place of Training- out-side country response is 77%?

7- Authors mentioned that Questioners were translated-and re-translated into Arabic as well, what was the reliability and validity of translation?

8- Reference from 1-9 and 10 to onwards there is formatting difference, need

9- p-value please write in internationally accepted format in Italic (p)font in result section.

Thanks & Regards.

Reviewer #2: The authors performed a study addressing “Fidelity of family centered care model to early disability diagnosis and rehabilitation in the United Arab Emirates”. However, there is one unclear area. In "Study participants" on page 7, why were the 252 health care professionals who agreed to participate in the study reduced to 150? Please elaborate on the reason. Also, if necessary, please provide a flowchart.

6. PLOS authors have the option to publish the peer review history of their article (what does this mean?). If published, this will include your full peer review and any attached files.

Reviewer #1: **Yes: **Prof Irshad Ahmad

Reviewer #2: No

---

## [Decision Letter · Decision Letter 1]

12 Mar 2024

Fidelity of family centered care model to early disability diagnosis and rehabilitation in the United Arab Emirates

PONE-D-24-03770R1

Dear Dr. Opoku,

We’re pleased to inform you that your manuscript has been judged scientifically suitable for publication and will be formally accepted for publication once it meets all outstanding technical requirements.

Kind regards,

Amin Nakhostin-Ansari

Academic Editor

PLOS ONE

Additional Editor Comments (optional):

Reviewers' comments:

Reviewer's Responses to Questions

**Comments to the Author**

1. If the authors have adequately addressed your comments raised in a previous round of review and you feel that this manuscript is now acceptable for publication, you may indicate that here to bypass the “Comments to the Author” section, enter your conflict of interest statement in the “Confidential to Editor” section, and submit your "Accept" recommendation.

Reviewer #1: All comments have been addressed

Reviewer #2: (No Response)

2. Is the manuscript technically sound, and do the data support the conclusions?

Reviewer #1: Yes

Reviewer #2: Partly

3. Has the statistical analysis been performed appropriately and rigorously? 

Reviewer #1: Yes

Reviewer #2: I Don't Know

4. Have the authors made all data underlying the findings in their manuscript fully available?

Reviewer #1: Yes

Reviewer #2: (No Response)

5. Is the manuscript presented in an intelligible fashion and written in standard English?

Reviewer #1: Yes

Reviewer #2: (No Response)

6. Review Comments to the Author

Reviewer #1: Thanks for addressing all the commentes. Now the manuscript is good for next step. I wisah all the best for all the authors.

Reviewer #2: (No Response)

7. PLOS authors have the option to publish the peer review history of their article (what does this mean?). If published, this will include your full peer review and any attached files.

Reviewer #1: No

Reviewer #2: No

---

## [Editor Report · Acceptance letter]

26 Mar 2024

PONE-D-24-03770R1 

PLOS ONE

Dear Dr. Opoku, 

I'm pleased to inform you that your manuscript has been deemed suitable for publication in PLOS ONE. Congratulations! Your manuscript is now being handed over to our production team.

Kind regards, 

on behalf of

Dr. Amin Nakhostin-Ansari 

Academic Editor

PLOS ONE